# When are Lemons Purple? The Concept Association Bias of Vision-Language Models

**Yingtian Tang**[*,♣]     **Yutaro Yamada**[*,◇]     **Yoyo Zhang**     **Ilker Yildirim**[◇]

♣ EPFL     ◇Yale University
yingtian.tang@epfl.ch, yutaro.yamada@yale.edu

## Abstract

Large-scale vision-language models such as CLIP have shown impressive performance on zero-shot image classification and image-to-text retrieval. However, such performance does not realize in tasks that require a finer-grained correspondence between vision and language, such as Visual Question Answering (VQA). As a potential cause of the difficulty of applying these models to VQA and similar tasks, we report an interesting phenomenon of vision-language models, which we call the Concept Association Bias (CAB). We find that models with CAB tend to treat input as a bag of concepts and attempt to fill in the other missing concept crossmodally, leading to an unexpected zero-shot prediction. We demonstrate CAB by showing that CLIP's zero-shot classification performance greatly suffers when there is a strong concept association between an object (e.g. eggplant) and an attribute (e.g. color purple). We also show that the strength of CAB predicts the performance on VQA. We observe that CAB is prevalent in vision-language models trained with contrastive losses, even when autoregressive losses are jointly employed. However, a model that solely relies on autoregressive loss seems to exhibit minimal or no signs of CAB.

## 1 Introduction

Recent large-scale vision-language models such as CLIP (Radford et al., 2021) and ALIGN (Jia et al., 2021) have shown remarkable performance on zero-shot classification and text-image retrieval tasks. These models are trained via cross-modal contrastive learning on web-scale image-text pairs and obtain powerful multimodal representations. Encouraged by these strong zero-shot capabilities, several recent papers explored CLIP for more complicated vision-language tasks. The initial attempt made by (Shen et al., 2022) reports near chance ac-

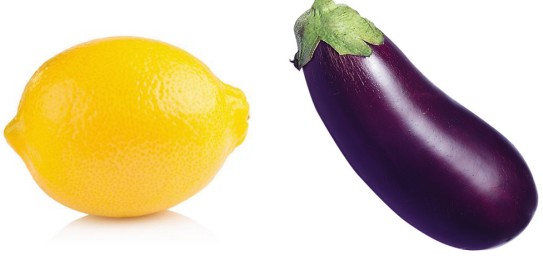

CLIP: "In this picture, the color of the lemon is purple."

Figure 1: When we ask CLIP the color of the lemon in the above image, CLIP answers "purple". The text prompt we use is "In this picture, the color of the lemon is [mask]", where CLIP picks one from [red, green, yellow, orange, purple].

curacy for zero-shot performance of CLIP on VQA-v2 (Goyal et al., 2017), a common visual question answering benchmark. However, they simply use "question: [question text] answer: [answer text]" as text input for the text encoder of CLIP, which makes the prediction harder than it should be. A subsequent work (Song et al., 2022) proposes a better prompt generation method. They convert questions into masked prompts (e.g. "What's in the bowl behind the cake" becomes "The [mask] is in the bowl behind the cake"), and filter impossible answers using a language model, which improves CLIP's zero-shot performance on VQA-v2.

However, the zero-shot performance of CLIP on VQA-v2 is still not state-of-the-art (Shen et al., 2022). In this paper, we report a phenomenon, which we call Concept Association Bias (CAB), as one of the reasons why CLIP struggles with VQA.

To describe this phenomenon, we present a simple image containing a "lemon" and an "eggplant" to CLIP, and ask what color the lemon is, as shown in Figure 1. Surprisingly, CLIP predicts "purple" with high confidence. When we instead ask for the color of the eggplant, CLIP answers "yellow". To cross-check this phenomenon, we formulate a binary zero-shot image classification task on the same

---

[*]Equal contribution.

image where the two labels are "yellow lemon" and "purple lemon", and find that CLIP predicts "purple lemon" with high confidence.

We hypothesize that this phenomenon comes from the discrepancy between what is described in the image and text input, where CLIP attempts to fill in the missing concept. The association between "purple" and "eggplant" is strong, so when asked to fill in the mask in "[mask] lemon", predicting "purple" instead of "yellow" makes more sense for CLIP, because the text description of "purple lemon" is aligned with the image that contains both a lemon and an eggplant more faithfully than "yellow lemon", which only describes the lemon in the image. In fact, when we randomize the color of the lemon and eggplant (e.g. "red" for the lemon and "green" for the eggplant), this bias disappears, and CLIP picks the color almost randomly between the two. We also find that CAB exists for more general object-attribute relationship such as the part-whole relationship (e.g. "humans" tend to have "clothes" on, and "trees" tend to have "leaves".)

Does CAB exist in other vision and language models as well? To answer this question, we also test BLIP (Li et al., 2022), BLIP-2 (Li et al., 2023), and OFA (Wang et al., 2022). We find that CAB exists in both BLIP and BLIP-2, but not in OFA, which is trained solely with autoregressive loss.

Finally, we demonstrate that enabling deeper interaction between modalities in CLIP can mitigate CAB. In particular, we show that extending CLIP with an additional Transformer layer on top and fine-tuning it on VQA is particularly helpful. Across such variants of CLIP, we report that the lower the degree of CAB, the higher a model performs on visual question answering. However, we also find that this fine-tuning method may not be a comprehensive solution for the more general binding problem (Greff et al., 2020), such as accurately connecting attribute and object representations, which leaves room for further research.

## 2 Related Work

**Vulnerability of vision and language models** There are a number of papers that study the robustness of vision and language models. Some prior work (Sinha et al., 2021) shows that Transformer trained via Masked Language Modeling (Devlin et al., 2019) is insensitive to word orders, suggesting that the success of BERT largely depends on learning higher-order word co-occurrence

rather than learning syntactic and semantic abstractions. Many benchmarks are proposed to evaluate robustness of ImageNet models towards various perturbations including common corruption (Hendrycks and Dietterich, 2019), image style change (Hendrycks et al., 2021), and different viewpoints (Barbu et al., 2019). Our work differs from these studies that are purely based on language or vision, because CAB is a cross-modal phenomenon, which occurs when both image and language data are used.

**Compositionality in vision and language models** The issue of vision and language models struggling with complex compositional questions has been studied before, where researchers have proposed enhanced training methods and modified architectures to tackle this problem (Basu et al., 2023; Nayak et al., 2022; Jiang et al., 2023). Bogin et al. (2021) tests compositional generalization of vision and language models. Thrush et al. (2022) introduced a probing dataset called Winoground, which evaluates visuo-linguistic compositionality of vision and language models. They evaluate a diverse range of state-of-the-art vision and language models, including CLIP, but all of them perform close to or below random chance. A subsequent work (Diwan et al., 2022) shows that Winoground requires not only compositional language understanding but also other abilities such as sophisticated common-sense reasoning and locating small objects in low resolution images, which most vision and language models currently lack. The work (Lewis et al., 2023) is the most relevant to our research, although it primarily deals with toy datasets. Our work also reveals brittleness of vision-language models through the lens of CAB, which has been overlooked in the past.

## 3 The Concept Association Bias

The zero-shot image classification of CLIP is remarkable for images that contain a single concept. However, when there are multiple concepts in the image but the text input does not cover all of them, the zero-shot classification of CLIP can be significantly biased towards the missing concept(s). We call this bias the Concept Association Bias (CAB). We first showcase this bias using color recognition tasks.[1] For this analysis, we use the Natural-Color

---

[1]For all experiments in the main text, we use the ResNet50-x4 backbone for CLIP. The results are consistent with the ViT backbone, which are included in the appendix.

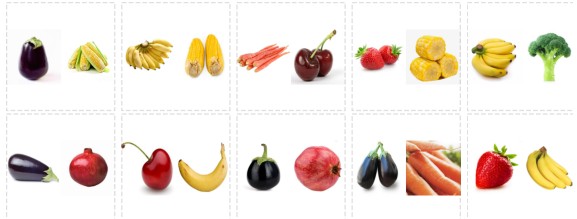

Figure 2: Example images from Natural-Color Dataset (NCD) (Anwar et al., 2022), modified for our color recognition tasks so that each image contains two different objects.

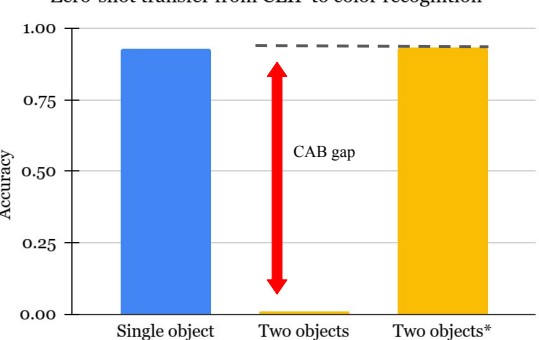

Figure 3: Zero-shot performance of CLIP on color recognition tasks using NCD (Anwar et al., 2022). CLIP achieves almost perfect accuracy when there is a single object in the image, but the accuracy significantly drops with two objects. "Two object*" refer to the case in which we instead measure the accuracy of predicting the color of the object B when it is asked for the color of the object A, where we see 80% zero-shot accuracy. We claim this gap between Two objects and Two objects* is a result of the Concept Association Bias (CAB).

Dataset (NCD) (Anwar et al., 2022), which is a dataset of vegetables and fruits with a white background. We take the following objects: *banana, brinjal, broccoli, carrot, cherry, corn, cucumber, lemon, orange, plum, pomegranate, strawberry, tomato*. We then randomly sample two images with different vegetable types and place the two objects side-by-side, resulting in 494 images in total. Examples are shown in Figure 2.

For zero-shot transfer from CLIP to our color recognition task, we ask for the color of one of the objects in the image. The labels we use are "red", "yellow", "purple", "green", and "orange", so it is a 5-way color recognition task. When there is a single object in the image, we use the following text prompt: "In this picture, the color of the object is [mask]." When there are two objects in the image, we specify one of these objects in the prompt. For example, if there is a lemon and another object in the image, the prompt takes the following format: "In this picture, the color of the lemon is [mask]."

The results are shown in Figure 3. We first note that the zero-shot performance of CLIP on our color recognition task is almost perfect when there is a single object per image ("Single object" in Figure 3). However, the classification performance degrades to below chance when there are two objects per image ("Two objects" in Figure 3).

How does this happen? We suggest that CLIP does not have a mechanism that stores object-centric representation that correctly binds the object's name and its attribute. In another words, CLIP processes its input as a "bag of concepts".

To inspect this possibility, we look at what kind of mistakes CLIP makes when there are two objects A and B. We find that many mistakes are derived from a common source. That is, when asked for the color of object A, CLIP often predicts the color of object B in the image. In fact, when we measure the accuracy of predicting the color of the object B

when in reality it is asked to predict the color of the object A, we see that the zero-shot transfer performance of CLIP is much higher ("Two objects*" in Figure 3), approaching the single object accuracy.

To understand this phenomenon, we find it helpful to consider two variables per object, where each variable represents the object's name in the image and the color attribute of the object, as shown in Figure 4. When the colors are natural (Figure 4 (a)), both the object "lemon" and its attribute "yellow" in the image are fully explained by the word "lemon" in the text prompt, resulting in the concept of the eggplant remaining. When CLIP performs zero-shot color recognition, we see that placing the color "purple" in the prompt can most faithfully explain the remaining concept of the eggplant in the image (Figure 4 (b)).

The above explanation suggests that there is a strong association between the color "purple" and the object "eggplant" in CLIP to the point where "purple" can partially explain the concept of the eggplant. What if we break this strong association? Does the gap between Two objects and Two objects* disappear?

To test this, we generate images of fruit and vegetable in unnatural color using Stable Diffusion 2.0 (Rombach et al., 2022) with a prompt format '[color name] [fruit/vegetable name]', and filter bad images by ourselves. Examples are shown in Figure 5. We call this dataset UNnatural-Color

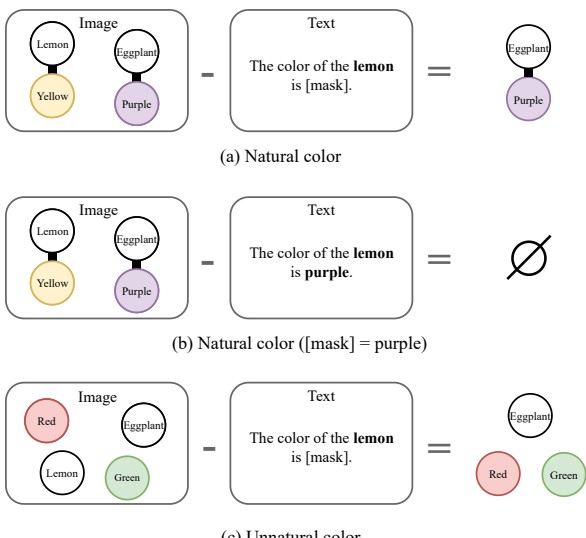

(a) Natural color

(b) Natural color ([mask] = purple)

(c) Unnatural color

Figure 4: The concept binding diagram. Two variables per object represent the object name and its attribute (e.g. color), respectively. We suggest that the text prompt and the image are represented as two separate "bags of concepts" in CLIP. When a pair of object-attribute concepts are naturally associated with each other, both concepts can be accounted for by including in the prompt either of the object or the attribute. When only some of the concepts in the image are included in the text, this leaves other concepts in the image unaccounted for.

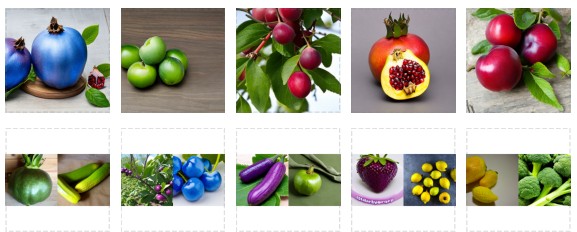

Figure 5: Examples from UNCD. Single object (Top) and Two objects per image (Bottom).

Dataset (UNCD). We repeat the same experiment on UNCD. The results are shown in Figure 6. We see that the zero-shot performance for a single object is still high, suggesting that CLIP can pick up the color attribute even if the color is not strongly associated with the object itself. However, for the two object cases, we see that there is almost no difference between Two objects and Two objects* tasks. In other words, CLIP predicts the two non-associated colors in the image with almost equal chance. We also create a version of NCD, which we call UNCD-v2, where we artificially change the color of each fruit and vegetable of NCD to non-associated color. As shown in Appendix, we see a similar pattern of CAB as UNCD.

Why does the CAB gap disappear when objects

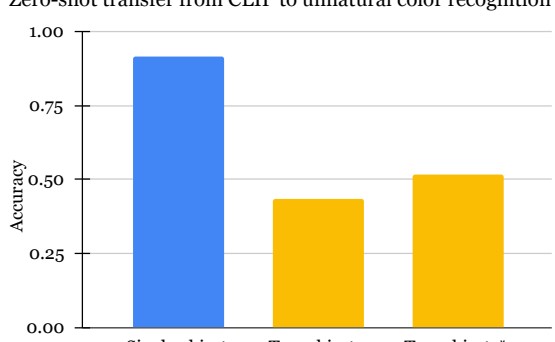

Zero-shot transfer from CLIP to unnatural color recognition

Figure 6: Zero-transfer performance of CLIP to color recognition on UNCD, where we assign non-associated color to each vegetable. CLIP achieves 80% accuracy when there is a single object in the image. While the accuracy drops for Two objects, the drop is not as significant as the NCD case. Furthermore, the gap between Two objects and Two objects* vanishes, compared to the NCD case.

are paired with random attributes in images? This result arises from a common mechanism that impacts both the Two objects and Two objects* tasks. To see this, we go back to our diagram in Figure 4 (c). When the colors are unnatural (*e.g.*, a lemon in red color and an eggplant in green color), then the remaining bag of concepts that are yet to be explained by the text include "red", "green", and "eggplant". This is because the color "red" is not associated with the concept of "lemon", and therefore the word "lemon" in the text prompt cannot explain the color "red", unlike the case that uses natural color. As a result, CLIP can choose either "red" or "green" for color recognition. And indeed, surprisingly, CLIP randomly chooses between the two – it does not associate the concept of "red" with the lemon, even though in the image the lemon unambiguously appears in red. Likewise, for the Two objects* task (in which the correct prediction is defined as the color of object B when asked for object A), CLIP essentially randomly picks one of the two colors present in the image, despite the fact that each object has their own very distinct color.

## 3.1 CAB exists on real-world dataset

So far, we use NCD to verify the existence of CAB. Here, we test CAB on a common visual question answering benchmark: VQA-v2 (Goyal et al., 2017). We perform the zero-shot transfer of CLIP to the color-related questions in VQA-v2, where our labels are *beige, black, blue, brown, green, gray, purple, red, orange, pink, white, yellow*, and *silver*.

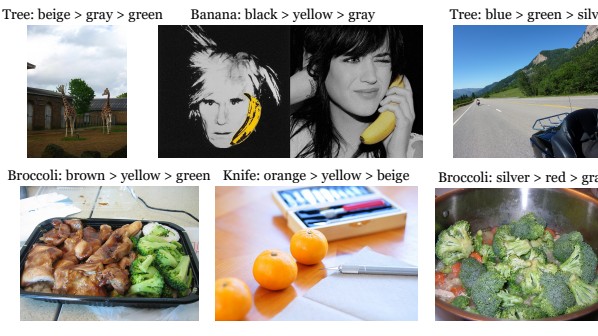

Tree: beige > gray > green    Banana: black > yellow > gray    Tree: blue > green > silver

Broccoli: brown > yellow > green    Knife: orange > yellow > beige    Broccoli: silver > red > gray

Figure 7: The Concept Association Bias (CAB) in VQA-v2. The text prompt is in the following format: "The color of the [object] is [color]." The first word on top of each image indicates the word used in place of [object] and the remaining color names are listed in the order CLIP chooses them for [color].

We use the prompt format "The color of the [object] is [color]." We show example images and the CLIP's zero-shot color predictions (top three, in decreasing order) in Figure 7. We can see that these mistakes are a result of CAB. For example, when we use the prompt format "The color of the banana is [color]" and the image that contains both banana and black-and-white portraits of people, CLIP answers "black" instead of "yellow". We randomly sample 100 mistakes CLIP makes out of all color-related questions, and manually inspect these images to identify if the mistakes are based on CAB. We find that roughly 50 mistakes are due to CAB. In Section 6, we illustrate how the degree of the CAB affects the performance on VQA-v2 in more detail.

### 3.2 CAB exists for attributes other than color

In this section, we test whether or not CAB exists for attributes beyond color. While there are various attributes we can evaluate, here we focus on part-whole attributes. Part-whole attributes are suitable for our CAB experiment because just like color, we can construct a syntactically reasonable prompt by finding two objects with a part-whole relationship. For example, "A tree has leaves" would be a good example prompt for our test, where the verb "has" indicates the part-whole relationship between the two objects in the sentence. To evaluate the performance of zero-shot transfer of CLIP on part-whole recognition, we use the Rel3D dataset (Goyal et al., 2020), which was originally proposed to test spatial relationship understanding of vision models. Rel3D consists of images of 3D scenes, where two objects with a particular spatial relationship are situated in each scene. Example images from Rel3D

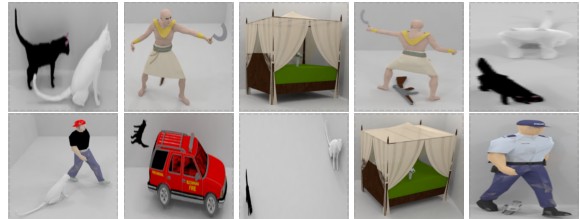

Figure 8: Example images from Rel3D (Goyal et al., 2020).

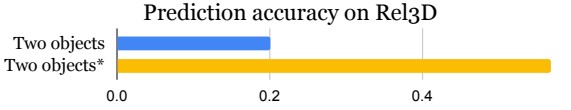

Prediction accuracy on Rel3D

Two objects
Two objects*

0.0    0.2    0.4

Figure 9: Zero-shot transfer performance of CLIP to part-whole recognition on Rel3D (Goyal et al., 2020). Similar to the color recognition task, CAB exists for part-whole recognition.

are shown in Figure 8.

We select 8 objects and their corresponding part attributes as follows: *(person, clothes), (camera, lens), (plant, leaves), (car, wheel), (cat, tail), (computer, screen), (bed, sheet), (gun, trigger)*. In total, we collect 356 images from Rel3D that contain one of these pairs. Prompt examples include "In this picture, the human has [mask]", "In this picture, the plant has [mask]" etc., and we let CLIP pick one from the 8 part attributes as above (e.g., clothes, lens, leaves, etc.).

The results are shown in Figure 9. We find that indeed, similar to the zero-shot color recognition task, the part-whole recognition task also shows CAB. This result suggests that CAB more generally applies to CLIP across types of object-attribute relations.

## 4 How does the strength of concept binding affect CAB?

In Section 3, we verify CAB for color recognition and part-whole recognition tasks. In this section, we investigate if varying the strength of the binding between two words affects the degree of CAB. We use ConceptNet (Speer et al., 2017) to measure the association strength between two words. Concept-Net is a knowledge graph that connects words with labelled, weighted edges. When selecting words, we focus on the edges that are labelled as "RelatedTo". For each Rel3D object name we used in Section 3.2, we pick 5 related words in the decreasing order of association strength, as shown in Table 1. For this concept recognition task, we use the following prompt format: "[object] [word]" and

| object | 1 | 2 | 3 | 4 | 5 |
|---|---|---|---|---|---|
| person | human | doll | character | statue | servant |
| camera | picture | flash | subject | photographer | tripod |
| plant | seed | tree | flower | green | cotton |
| car | drive | vehicle | motor | automobile | wheels |
| cat | feline | animal | pet | kitten | dog |
| computer | apple | desk | print | dell | data |
| bed | sleeping | furniture | mattress | place | pillows |
| gun | bullet | weapon | rifle | shooting | pistol |

Table 1: Object names from Rel3D (the first column) and the top five related words from ConceptNet (Speer et al., 2017). The smaller the column number is, the stronger the association to the corresponding object name is.

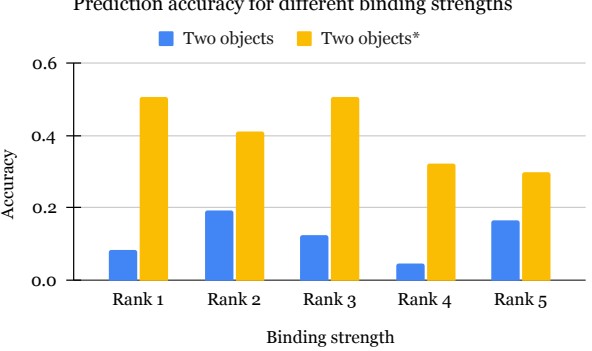

Figure 10: We vary the strength of concept binding, and compute the accuracy for Two objects and Two objects*. We see that as the association strength gets weaker, CAB becomes smaller, although it is somewhat noisy.

the label sets are restricted to the words in the same column. The results are shown in Figure 10. While it is noisy, we see that as the concept association becomes weaker, CAB becomes smaller.

## 5 CAB widely exists in vision-language models

In Section 3, we demonstrate CAB using CLIP. In this section, we extend our experiments to other vision-language models, including BLIP (Li et al., 2022), BLIP-2 (Li et al., 2023), and OFA (Wang et al., 2022).

BLIP and BLIP-2 are both multi-modal, multi-task model that has three heads and pre-training objectives: the CLIP-like image-text contrastive loss, the binary image-text matching loss that classifies whether the text corresponds to the image, and the language modeling loss, which autoregressively generates the caption for an image. For BLIP-2, there is the second stage of pre-training for visual conditioned language generation, bootstrapped from pre-trained LLM. In our experiment,

we treat these models with different heads separately and abbreviate them as "contrast", "match", and "caption". OFA unifies various vision and language tasks via a simple sequence-to-sequence framework and employs autoregressive loss for its objective.

For comparison across models, we define the CAB score as:

$$\frac{Acc_{two\ object*} - Acc_{two\ object} + 1}{2}$$

where $Acc$ stands for accuracy. The CAB score ranges from 0 to 1, and a higher score indicates a more severe CAB.

In Table 2, we report results on NCD for the aforementioned models and order them according to the CAB scores. While these networks all have high single-object recognition performance, they demonstrate a spectrum of levels of CAB.

The networks trained with contrastive or matching losses (i.e. CLIP, BLIP-contrast/match, BLIP-2-contrast/match) have stronger CAB than those trained with autoregressive loss (i.e. BLIP-caption, BLIP-2-caption, and OFA). Moreover, in comparison with BLIP, BLIP-2 uses a large language model as the final decoder on top of its sub-networks, making them more associated with the autoregressive loss and having less CAB scores than their counterparts in BLIP.

Furthermore, we observe that matching losses have lower CAB than contrastive losses for both BLIP and BLIP-2. Although the two losses are similar, the matching task uses cross-attention that jointly processes texts and images, whereas the contrastive task only uses unimodal self-attention. Based on this observation, we hypothesize that the deeper cross-modal interaction helps mitigate the CAB. We further investigate this in Section 6.

It is worth noting that the amount of CAB in autoregressive models is substantial. For example, Two objects*'s accuracy for BLIP-caption, BLIP-2-caption, and BLIP-2-FlanT5 are 0.471, 0.483, and 0.377 respectively. This means that when joitnly trained with contrastive losses, even autoregressive models are biased by concept association, resulting in Two objects*'s accuracy being much higher than random chance. The only model with minimal or almost no CAB is OFA, which sorely relies on autoregressive loss.

In Table 3, we compare CAB scores for CLIP with vision encoders in different sizes. Although

| Models | Two objects | Two objects* | Single | CAB |
|---|---|---|---|---|
| CLIP | 0.011 | 0.932 | 0.929 | 0.961 |
| BLIP-contrast | 0.086 | 0.879 | 0.846 | 0.896 |
| BLIP-match | 0.123 | 0.841 | 0.925 | 0.859 |
| BLIP-2-contrast | 0.138 | 0.840 | 0.844 | 0.851 |
| BLIP-2-match | 0.330 | 0.627 | 0.925 | 0.648 |
| BLIP-2-caption | 0.359 | 0.558 | 0.775 | 0.599 |
| BLIP-caption | 0.438 | 0.471 | 0.862 | 0.516 |
| BLIP-2-FlanT5 | 0.604 | 0.377 | 0.984 | 0.386 |
| OFA | 0.855 | 0.078 | 0.879 | 0.111 |

Table 2: CAB experiments using Natural Colorful Dataset (NCD) on multiple architectural variants. FlanT5 refers to BLIP-2-ViTg-FlanT5$_{XL}$.

| CLIP | Two objects | Two objects* | CAB |
|---|---|---|---|
| ViT-B/32 | 0.023 | 0.944 | 0.961 |
| ViT-B/16 | 0.059 | 0.886 | 0.913 |
| ViT-L/14 | 0.057 | 0.918 | 0.931 |
| ViT-L/14@336 | 0.058 | 0.926 | 0.934 |
| RN50 | 0.011 | 0.932 | 0.961 |
| RN50x4 | 0.045 | 0.916 | 0.936 |
| RN50x16 | 0.121 | 0.842 | 0.860 |
| RN50x64 | 0.074 | 0.872 | 0.899 |
| RN101 | 0.034 | 0.944 | 0.955 |

Table 3: CAB experiments for CLIP with vision encoders in different sizes.

the size varies greatly, the CAB score stays around the same level for these different CLIP models.

In Table 4 and Table 5, we compare CAB with Winoground (Thrush et al., 2022), whose task is to correctly match two images with two captions, but the key aspect is that both captions consist of the exact same words, albeit arranged differently. We compare CLIP, BLIP-contrast/match, and BLIP-2-contrast/match because they are more suitable for the Winoground matching task. We roughly see that as CAB decreases, Winoground performance goes up, which is aligned with what CAB attempts to measure. However, we also observe that using matching loss benefits more for both CAB and Winoground, presumably because the matching loss uses cross-attention between text and image encoders.

## 6 How can we mitigate CAB?

**Fine-tuning helps reduce CAB**    In the last section, we show that CAB can be seen in vision-language models, and it is especially prominent for purely contrastively trained models. In this section, we test our hypothesis from Section 5 that a deeper modality interaction helps mitigate CAB in

| Contrastive-loss network | CAB | Winoground-group |
|---|---|---|
| CLIP | 0.961 | 0.0724 |
| BLIP-contrast | 0.896 | 0.0800 |
| BLIP-2-contrast | 0.851 | 0.0850 |

Table 4: CAB vs. Winoground with contrastive models.

| Matching-loss network | CAB | Winoground-group |
|---|---|---|
| BLIP-match | 0.859 | 0.206 |
| BLIP-2-match | 0.648 | 0.235 |

Table 5: CAB vs. Winoground with matching models.

a controlled experiment using CLIP.

The idea of using deep modality interaction on top of image and text embeddings has been explored before in (Kim et al., 2021). However, in (Kim et al., 2021), the image and text encoders are shallow unlike CLIP. In (Shen et al., 2022), instead of using CLIP as is, they employ the architecture that uses the image encoder of CLIP, the BERT text embeddings, and a Transformer as an additional modality interaction module. They apply this model for vision and language tasks such as Visual Question Answering, Visual Entailment, and V&L Navigation tasks. The goal of (Shen et al., 2022) was to demonstrate that the image encoder of CLIP is more helpful than ImageNet-pretrained image encoders. For our architecture, we use both image and text encoders from CLIP, and also a Transformer for modality interaction on top of CLIP, as shown in Figure 11.

We conducted experiments in two settings: 1. Freezing CLIP and only fine-tuning the Transformer head, and 2. Fine-tuning both CLIP and the Transformer head. Following (Shen et al., 2022), we use VQA-v2 to fine-tune these model variants. We follow the standard pre-processing of VQA-v2, where we filter less common answers and select

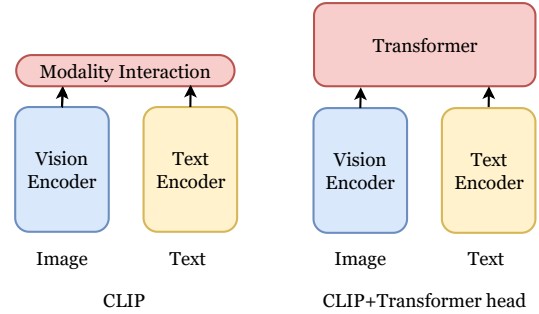

Figure 11: Original architecture of CLIP (Left) and the architecture we use for fine-tuning on VQA-v2 (Right).

| NCD | Two objects | Two objects* | CAB Score | VQA-v2 |
|---|---|---|---|---|
| CLIP(Frozen) | 0.352 | 0.094 | 0.371 | 0.542 |
| CLIP(Finetuned) | 0.328 | 0.168 | 0.420 | 0.390 |

Table 6: Given two objects A and B per image, when we ask the color of object A, "Two objects" refer to the case where we use the color of object A as true labels, and "Two objects*" refer to the case where we use the color of object B as true labels. For both cases, the Transformer head on top of CLIP are fine-tuned. See text for details of the two models.

3,129 answer vocabularies. We then formulate the VQA task as a classification problem over 3,129 answer vocabularies given images and questions as input. After fine-tuning on VQA-v2, we perform zero-shot color recognition using NCD to evaluate CAB in a similar manner to Section 3. That is, given two fruits in an image, we ask the color of one of the fruits. The results are shown in Table 6. We can see that adding deeper modality interaction reduces CAB (See Fig. 3 for comparison). Moreover, we also see that between these model variants, the lower the CAB score is, the higher the accuracy on VQA-v2 is. Does this relationship hold more generally? To see this, we prepare three other baselines: A. CLIP image encoder + BERT text embeddings + Transformer head, fine-tuned altogether; B. The same as A. but with the CLIP image encoder frozen; C. The same as A. but uses V&L pre-trained weights before fine-tuning. These architectures are based on (Shen et al., 2022).

V&L pre-training used the aggregated data from MS COCO Captions (Chen et al., 2015), Visual Genome Captions (Krishna et al., 2017), VQA (Antol et al., 2015), GQA (Hudson and Manning, 2019), and Visual7W (Zhu et al., 2016), which results in 9.18M image-text pairs. For C. we used the publicly available pre-trained model released by (Shen et al., 2022). For A. and B., we train the models on our own. We detail the hyperparameters in the appendix. The results are shown in Figure 12. We see that as the CAB Score becomes lower (i.e., as models become less susceptible to the concept association bias), the accuracy on VQA-v2 increases. This encourages us to reduce the bias derived from Concept Association to further improve vision and language models in general.

**Fine-tuning alone may not necessarily solve the binding problem**    The last section demonstrates that by introducing deeper modality interaction and fine-tuning, we can mitigate CAB on NCD. Can

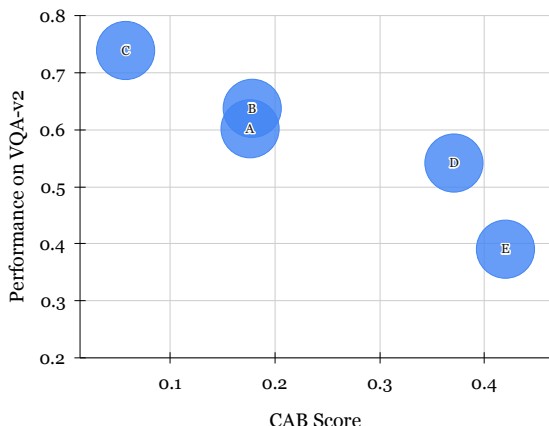

Figure 12: The lower the CAB Score (less susceptible to the Concept Association Bias), the higher the models perform on VQA-v2. A-E refers to different model configurations. A-C are detailed in the main text. D and E are the same as CLIP(Frozen) and CLIP(Finetuned) in Table 6, respectively.

such a procedure solve the binding problem (Greff et al., 2020) more generally? The binding problem for neural networks refers to the inability of models to dynamically bind information that is distributed across the network (Greff et al., 2020). In fact, we can view CAB as an instance of the binding problem because one of the causes of CAB is the model's inability to correctly bind object and attribute representations across modalities. If allowing for deeper modality interaction and fine-tuning helps to more faithfully bind attributes and objects across vision and language inputs, then we should expect to see accuracy improvement for the Two objects setting, even on UNCD. This is because a model that can successfully localize an object and its attributes, and separate the representation of different objects, should be able to identify the color of the queried object even if the object is randomly colored. Contrary to this prediction, we find that the accuracy on Two objects for these fine-tuned models is *lower* than the original CLIP, except for the model C, which uses large pre-training datasets. In fact, we find that these fine-tuned models also have lower accuracy for Two objects* (Table 7), indicating that the fine-tuned models most often choose a color that is not present in the image. This suggests that fine-tuning on VQA-v2 simply allows the model to pick up real-world co-occurrence of colors and objects more easily and consistently. Therefore, fine-tuning the Transformer head may not fundamentally solve the problem of correctly binding objects and their attributes.

| UNCD | CLIP(Original) | CLIP(Frozen) | CLIP(Finetuned) | A | B | C |
|---|---|---|---|---|---|---|
| Two objects | 0.436 | 0.216 | 0.216 | 0.308 | 0.313 | 0.574 |
| Two objects* | 0.517 | 0.077 | 0.132 | 0.165 | 0.153 | 0.105 |

Table 7: The performance of CLIP with fine-tuned deeper interaction module. We see that fine-tuning rather mostly harms the accuracy for Two objects, instead of improving it, which suggests that fine-tuning may not solve the more general binding problem (Greff et al., 2020). The values for CLIP(Original) are the same as Figure 6. A-C refer to specific fine-tuned model configurations, detailed in the main text.

## 7 Conclusion

Every object has a set of concepts that are roughly associated with it. For instance, the object "lemon" can be associated with "yellow", "fruit", and so on. Such concept association is automatically learned in vision-language models, to the point where the word "yellow" can partially explain the object "lemon" in certain cases. We establish that the Concept Association Bias (CAB) exists for vision-language models through a series of experiments, and find that the models trained with contrastive loss are especially affected. Furthermore, we verify that the lower the degree of CAB is, the higher the performance of VQA-v2 is. Contrastive models like CLIP is increasingly popular in both computer vision and natural language processing. We hope our work raises awareness of the brittleness of contrastive objectives as we develop new vision and language models.

## Limitations

While we verify CAB in zero-shot transfer of contrastive based vision and language models for color recognition and part-whole recognition, there are other object-attribute relationships that are not explored in this paper such as object material, shape, and texture. Additionally, we focus our study on VQA, as the task format of VQA is directly applicable to our CAB experiments. An interesting future study to complement our work is to explore the effect of CAB on other vision-language tasks (e.g., visual entailment (Song et al., 2022)), and explore other methods to mitigate CAB. Finally, we focus on CLIP, BLIP, BLIP-2, and OFA in this study. Future work should also investigate other vision-language models that incorporate more extensive modality interactions (e.g., FLAVA (Singh et al., 2022) and ALBEF (Li et al., 2021). However, given that these models are widely adopted in both computer vision and natural language processing

for a wide variety of downstream tasks, we believe our results are important to the community.

## Ethics statement

Although our findings may not have immediate implications for the misuse of AI systems, it is essential to acknowledge that biases and unintended behaviors exhibited by models like CAB can pose potential risks, including social bias and other forms of harm. Addressing these biases and limitations becomes imperative to ensure the ethical and fair utilization of vision-language models in real-world scenarios. We strongly advocate for continued research and development that emphasizes transparency, fairness, and accountability in the design and implementation of vision-language models.

## Acknowledgements

YY's research is partially supported by Masason Foundation.

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

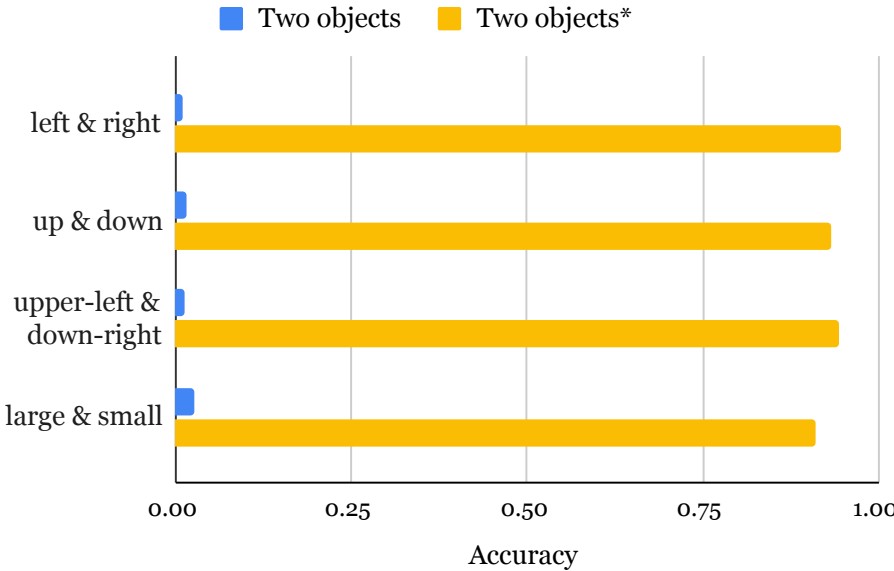

Figure 13: The Concept Association Bias (CAB) remains regardless of the spatial configurations such as "left & right", "up & down", "upper-left & down-right", and "large & small". We use the same subset of NCD as in Figure 3.

## A    Details of UNCD

We used '512-base-ema.ckpt' from Stable Diffusion's official github repository[2]. We used the deafult hyperparameters for sampling, except we set the size of images to be 512 by 512, and we sample 4 images for each prompt. We used the following negative prompt: 'low-res, oversaturated, ugly, cartoon, grain, out of focus, ambiguous, blurred, split frame, out of frame, cropped, multiple frame, split panel, multi panel, people, human, logo'

## B    CAB is robust across different conditions

### B.1    The spatial arrangement has almost no effect on CAB

In our earlier experiments on NCD and UNCD, two objects are positioned side-by-side. To see if CAB is robust to the positioning of objects, we vary the spatial arrangement of the two objects in the image. Concretely, we test the following spatial configurations: left & right, up & down, and upper-left & down-right. We also vary the size of the two objects for left & right, which is denoted as "large & small". As Figure 13 shows, CAB is not affected by either spatial arrangements or the object size.

### B.2    CAB persists for prompt variations

The original CLIP paper (Radford et al., 2021) reports that varying text prompts changes zero-shot transfer performance of CLIP. Here, we test if CAB remains effective when we vary the prompt. As Figure 14 shows, CAB is relatively stable across prompt variations. It is interesting that as long as the names of the object and color (denoted as [object] and [color]) are included in the prompt, CAB exists even if text prompts are semantically meaningless such as: "[object] [color]", "The color of [color] is [object]" and "This prompt is random [object] [color]". Moreover, even if we negate the sentence (e.g. "The color of [object] is not [color]") CAB still exists, which suggests that the text encoder of CLIP seems to ignore the negation.

---

[2]https://github.com/CompVis/stable-diffusion

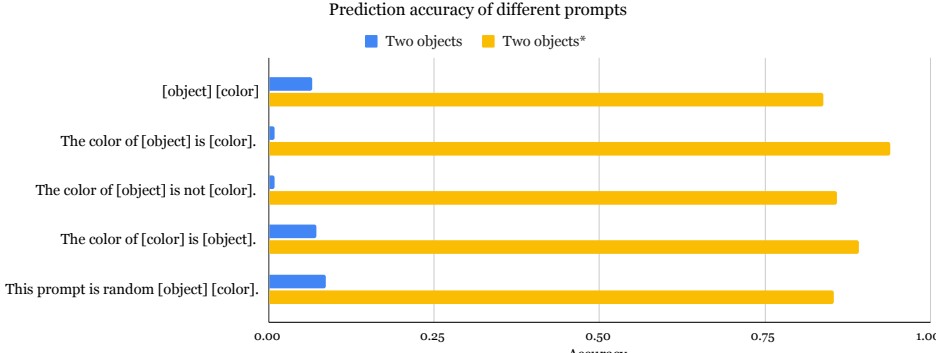

Figure 14: The Concept Association Bias (CAB) is relatively stable across prompt variations, including semantically meaningless prompts. We use the same subset of NCD as in Figure 3

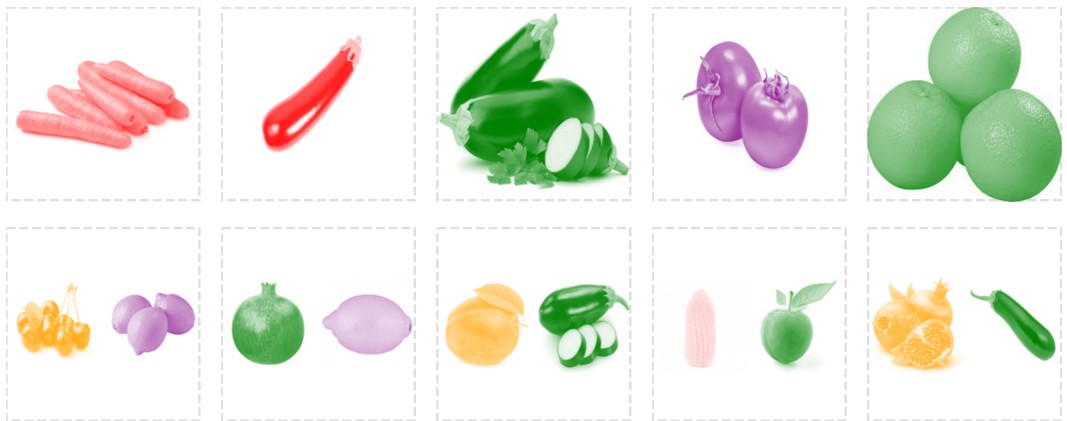

Figure 15: Examples from UNCD-v2. Single object (Top) and Two objects per image (Bottom).

## C   CAB experiments using UNCD-v2

We utilize the same instances of NCD, but we assign non-associated colors to each vegetable. Figure 15 displays some example images. In Figure 16, we observe that Two objects and Two objects* exhibit similar performance, mirroring the pattern observed with UNCD discussed in the main text.

## D   Inspecting object representations of CLIP

The section 6 demonstrates that by introducing more modality interaction, we can mitigate CAB. However, this requires an additional procedure to fine-tune the newly introduced module. Can we alleviate CAB by spatially pooling features of the original CLIP? Such an approach would work if CLIP develops localized object-centric representations. To investigate this possibility, we use NCD and take the image tokens that correspond to the left half of images ("Left Pool"), and conduct zero-shot classification to predict the color of the left object of an image. We compare the performance of this procedure with the case where we use the average of all tokens in the image ("Global Pool"). If CLIP develops localized object-level representations, we should see an increase in accuracy compared to Global Pool. However, as shown in Table 8, we see that the accuracy of the left pooling is lower than global pooling. (We also show the accuracy values based on the original CLIP CLS embeddings.) This suggests that the features of an object that is positioned in the left half of the input image are propagated to the right half as well, so taking the features strictly from the left side reduces the information that is necessary to accurately predict the color. Therefore, when there are multiple objects, we can see that CLIP struggles with the binding between object representations and attribute representations. Future work should explore incorporating object-centric learning into CLIP to guide and structure the binding between objects and their attributes.

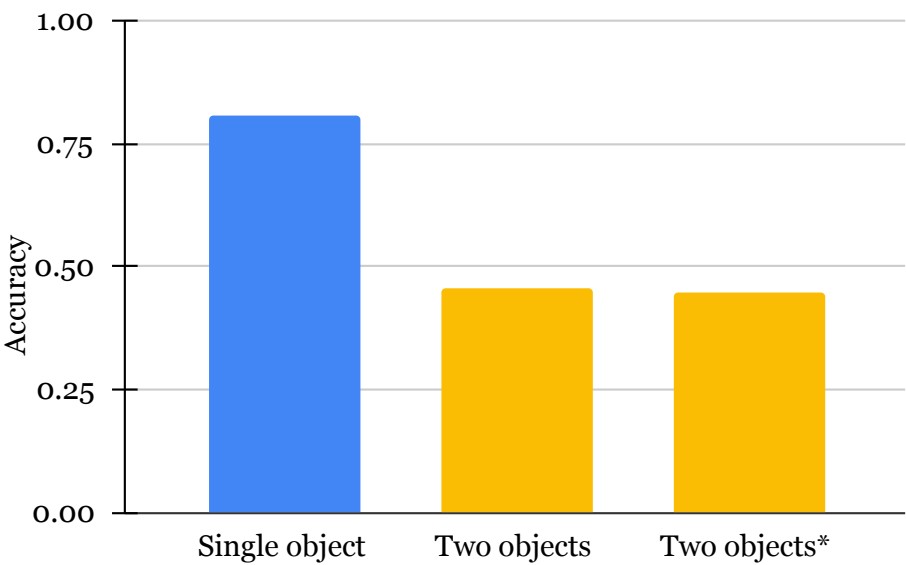

Figure 16: Zero-transfer performance of CLIP to color recognition on UNCD-v2, where we assign non-associated color to each vegetable. CLIP achieves 80% accuracy when there is a single object in the image. While the accuracy drops for Two objects, the drop is not as significant as the NCD case. Furthermore, the gap between Two objects and Two objects* vanishes, compared to the NCD case.

|  | CLS | Global Pool | Left Pool |
|---|---|---|---|
| NCD | 0.617 | 0.417 | 0.209 |
| UNCD | 0.474 | 0.429 | 0.134 |

Table 8: 5-way color classification accuracy using spatial pooling. The prompt format we use is "The color is [mask]." The original CLIP uses the CLS embedding to compute the similarity between image and text embedding. Global Pool takes the average of image tokens as their image embedding. Left Pool take the average of the tokens corresponding to the left side of the image. The reason why CLS for NCD is higher than 0.5 is that sometimes the color of two vegetables in the image are the same. In that case, the zero-shot classifier's prediction is almost always correct, which biases the overall accuracy towards 1.0.

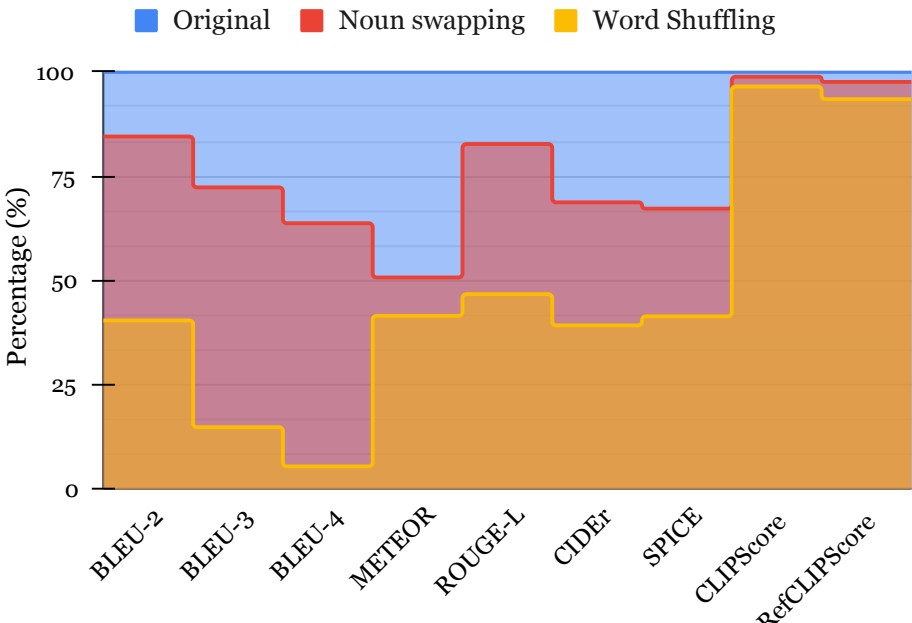

Figure 17: Ratio of caption score before and after text manipulation. We can see that CLIPScore operates as a bag of concepts, which is not affected by either noun swapping or word shuffling, compared to other caption metrics.

## E    Hyperparameter details for fine-tuning

In our experiment of fine-tuning the modality interaction architecture (Section 6), we used the exact same hyperparameters as those in (Shen et al., 2022). The specific repository we used is `https://github.com/clip-vil/CLIP-ViL/tree/master/CLIP-ViL-Pretrain`. They originally used a BERT text encoder and CLIP image encoder. When replacing the BERT text encoder with CLIP text encoder (*i.e.,* the cases D and E in Figure 12), we add a linear layer on top of the CLIP text encoder to map the embedding dimension to be the same as the Transformer head. For all fine-tuning experiments, we trained for 5 epochs, which is the default number of epochs in (Shen et al., 2022).

## F    CLIPScore experiment: CAB suggests precaution in downstream applications of CLIP

As we see in Section 3, CLIP tends to treat input as a bag of concepts, which can have undesired consequences when we use CLIP for downstream tasks. Here, we examine a recent application of CLIP, and find that it suffers from this phenomenon. CLIPScore (Hessel et al., 2021) was recently proposed as a way to assess the quality of image captioning models. In contrast to reference-based scores, which compare the similarity between generated captions and reference captions, CLIPScore simply compares the similarity between the embedding from input images and the embedding from corresponding captions generated by an image captioning model. The original paper reports high similarity of CLIPScore to human judgement, which is one of the favorable attributes of CLIPScore compared to existing reference-based captioning scores. Here, we show that CLIPScore can be *insensitive* to swapping and shuffling of words in a sentence, although humans can easily tell such differences.

We first use NLTK (Bird et al., 2009) to find part-of-speech tagging, and randomly shuffle the nouns within each sentence ("noun swapping"). We also prepare a baseline where we shuffle all words in each sentence ("word shuffling"). We then evaluate the ground truth captions of the validation set of VQA-v2 (Goyal et al., 2017) using CLIPScore, and compare how our shuffling procedures affect CLIPScore. In Figure 17, we see that there is almost no effect of our text permutations on CLIPScore and its reference augmented version RefCLIPScore, while other reference-based score methods are affected. For example, given a sentence that reads "A man is walking into the room", CLIPScore returns almost the same score for "A room is walking into the man." This further illustrates that CLIP treats the input sentence as a bag of concepts, and calls for caution when we use CLIPScore to evaluate image captioning models.

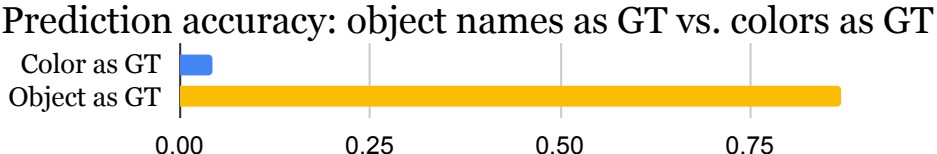

Figure 18: When both color and object names are available for ground truth labels, CLIP tends to pick object names over colors, suggesting that they are not completely interchangeable.

## G   Are the object name and attribute interchangeable?

We see evidence that the word "purple" serves as a replacement for the word "eggplant" in our CAB experiments so far, which leads to a caption such as "purple lemon" to represent an image of a lemon and an eggplant. However, it would be especially surprising if the color attribute is completely interchangeable with the object name. To test this, we expand the labels we use for evaluating zero-shot transfer performance of CLIP. Previously, for color recognition task, we use the colors as our labels. We now include the object names as our labels in addition to the color labels. In particular, we expand the label set from *yellow, red, ... , purple* to *yellow, red, ... , purple, banana, tomato, ... , eggplant*. Therefore, if the object names and color attributes are not completely interchangeable and the object names are more suitable to explain the image than attributes, then we should see a decrease in accuracy when we use the colors as our ground truth (GT) label. The results are shown in Figure 18. We see that the accuracy for "Color as GT" is much lower than "Object as GT", which suggests that CLIP only uses colors when object names are not available.

## H   Additional related work

**Peculiarities of CLIP**   In the image generation community, it has been reported that state-of-the-art models such as DALL·E 2 (Ramesh et al., 2022) struggle with compositionality (Rassin et al., 2022). One of the potential causes of such failure has been attributed to the use of CLIP-based image encoder (Ramesh et al., 2022). In fact, image generation models that do not use CLIP such as Imagen and Parti are known to be better at generating images that require compositional reasoning (Saharia et al., 2022; Yu et al., 2022). However, few works go into depth to analyze the behavior of CLIP in zero-shot image classification and visual question answering. Our analysis based on CAB offers a new perspective on the weakness of CLIP-based models for compositional reasoning.