# OpenReview forum: "When are Lemons Purple? The Concept Association Bias of Vision-Language Models"
_EMNLP/2023/Conference — EMNLP 2023 Main_

### Official Review · Reviewer_9Cqi · 2023-08-02

**Soundness:** 4

**Excitement:**

4: Strong: This paper deepens the understanding of some phenomenon or lowers the barriers to an existing research direction.

**Paper Topic And Main Contributions:**

This paper deeply investigates the Concept Association Bias (CAB) in the task of VQA. Through many experimental results, this paper finds that the models with CAB tend to treat input as a bag of concepts and attempt to fill in the other missing concept. This paper also observes that CAB is prevalent in vision-language models trained with contrastive losses.

**Questions For The Authors:**

Question A: As described in line 412 - 414, `the CAB score stays around the same level for these different CLIP models`, what does `the same level` means? The scores range from 0.860 to 0.961(the difference is more than 0.1) are around the same level?

**Reasons To Accept:**

1. Enough experiments and many inspiring findings. The Concept Association Bias did not be scrutinized carefully.
2. Good writing. The structure of the paper is clear and reading it is smooth and natural.

**Reasons To Reject:**

1. Some of the observations can be more solid. For example, as described in line 493 - 495, the relation between the CAB Score and the accuracy can be quantified statistically whether the correlation is significant enough.


**Reproducibility:**

4: Could mostly reproduce the results, but there may be some variation because of sample variance or minor variations in their interpretation of the protocol or method.

**Reviewer Confidence:**

3: Pretty sure, but there's a chance I missed something. Although I have a good feel for this area in general, I did not carefully check the paper's details, e.g., the math, experimental design, or novelty.

---

> ### Author Rebuttal · Authors · 2023-08-28
>
> We are glad that the reviewer finds our work inspiring.
>
> - *Some of the observations can be more solid. For example, as described in line 493 - 495, the relation between the CAB Score and the accuracy can be quantified statistically whether the correlation is significant enough*
>
> Thank you for this suggestion. We calculated Pearson's correlation coefficient between VQA-v2 scores and CAB scores and its corresponding p-value (line 493-495), and obtained the following significant relationship: Pearson's correlation coefficient r=-0.94 and p-value=0.017 (statistically significant under the conventional cut-off of .05). We'll include this in the text.
>
> - *As described in line 412-414, the CAB score stays around the same level for these different CLIP models, what does the same level means? The scores range from 0.860 to 0.961(the difference is more than 0.1) are around the same level?*
>
> What we mean by "the same level" is that the variability of CAB scores in Table 3 is in a similar range (0.860 to 0.961), compared to the variability of CAB scores in Table 2 (0.111 to 0.961), which highlights that the CAB depends more on training objectives & model architectures rather than model sizes. Thank you for pointing this out; we will clarify this in the revision.

---

### Official Review · Reviewer_ftk9 · 2023-08-03

**Soundness:** 3

**Excitement:**

3: Ambivalent: It has merits (e.g., it reports state-of-the-art results, the idea is nice), but there are key weaknesses (e.g., it describes incremental work), and it can significantly benefit from another round of revision. However, I won't object to accepting it if my co-reviewers champion it.

**Paper Topic And Main Contributions:**

This paper find that CLIP’s zero-shot classification performance suffers when there is a strong concept association between an object and an attribute. Multimodal vision and language models as well such as BLIP, BLIP-2, and OFA are evaluated and it is observed that models solely relies on autoregressive loss exhibit minimal or no signs of CAB.

**Reasons To Accept:**

The compositionality and attributed binding of VL models is an important problem. This paper systematically studies this problem and gives some insights.

A comprehensive evaluation of different popular VL models ranging from models based on CE loss to models based on regressive loss, is performed.

The experimental parts are solid and CLIP with different encoders are evaluated.

**Reasons To Reject:**

There is no new effective method proposed to address the Concept Association Bias problem even though the work did some study of the effect of finetuning.

The novelty is limited and many similar works have been done previously, e.g. [1][2][3][4].

[1] Learning to Compose Soft Prompts for Compositional Zero-Shot Learning

[2] Training-Free Compositional Image and Text Matching

[3] Does CLIP Bind Concepts? Probing Compositionality in Large Image Models

[4] Augmenting CLIP with Improved Visio-Linguistic Reasoning

**Reproducibility:**

3: Could reproduce the results with some difficulty. The settings of parameters are underspecified or subjectively determined; the training/evaluation data are not widely available.

**Reviewer Confidence:**

3: Pretty sure, but there's a chance I missed something. Although I have a good feel for this area in general, I did not carefully check the paper's details, e.g., the math, experimental design, or novelty.

---

> ### Author Rebuttal · Authors · 2023-08-28
>
> We are glad that the reviewer finds the problem important and our experiments solid.
>
> Thank you for suggesting these papers. We’ll make sure to discuss these in the revision. However, we would like to respectfully point out that the focus of our paper is not to propose new algorithms to solve the general problem of compositionality, but rather to provide new insights by reporting the interesting phenomenon of Concept Association Bias (CAB) as one of the reasons behind the frequently observed compositionality issues. In this way, our work aims to deepen our understanding of the source of compositionality issues. CAB targets a specific class of compositionality issues in existing V&L models. Indeed, none of the referenced papers [1-4] explicitly discusses Concept Association (e.g. “lemon” and “yellow”) as one of the reasons behind the compositionality issues of V&L models. To our knowledge, ours is the first work that points out this phenomenon. In addition, we have detailed studies of the properties of CAB, which provide the following insights:
> - We contrast the results from NCD and UNCD to have a more detailed understanding of the binding mechanism in CLIP; we also used ConceptNet to illustrate that the bias is amplified as the binding strength / concept association increases.
> - We illustrate that the CAB problem is not trivial by studying to what extent simple finetuning can fix it;
> - We study the correlation between CAB and contrastive & autoregressive objectives, which provides a better guide of designing objectives for large VL models and raises the caution when applying them to downstream applications.

---

### Official Review · Reviewer_JUDd · 2023-08-11

**Typos Grammar Style And Presentation Improvements:** 1. L096
**Soundness:** 4

**Excitement:**

4: Strong: This paper deepens the understanding of some phenomenon or lowers the barriers to an existing research direction.

**Paper Topic And Main Contributions:**

The paper investigates one important reason for the zero-shot failure of Vision Language models on VQA tasks. The main contributions of this paper can be summarized as follows:
1. The paper identifies an important problem of VL models, specifically on compositional questions in VQA. They identify that VL models tend to treat each image + description as two separate "bags of concepts". While they start with CLIP, they experiment with various models and find this to be true in general.
2. They attribute this to the contrastive loss training of the VL models.
3.  They provide a fine-tuning-based solution to mitigate this problem.

**Questions For The Authors:**

I had trouble understanding why the models were bad only on the compositional questions and not the individual ones. Can you please clarify that?

**Reasons To Accept:**

I think the paper asks a focused interesting question, exhaustively investigates that, proposes a mitigation mechanism, and clearly points out the limitations of the proposed solution. I like the expository style of writing.

**Reasons To Reject:**

I don't see many reasons.

**Reproducibility:**

5: Could easily reproduce the results.

**Reviewer Confidence:**

2: Willing to defend my evaluation, but it is fairly likely that I missed some details, didn't understand some central points, or can't be sure about the novelty of the work.

---

> ### Author Rebuttal · Authors · 2023-08-28
>
> We are glad that the reviewer finds our work important and interesting. We’ll increase the figure sizes and fix typos in the revision.
>
> *Q. Why the models were bad only on the compositional questions and not the individual ones. Can you please clarify that?*
>
> Thank you for a great question! For example, for a setup where we have an image of “purple banana” and nothing else, and ask for the color of banana ("the color of the banana is [mask]"), the reason why the model correctly responds as “purple” is that in the image, both the “purple” concept and the “banana” concept are present, and the query text that includes both the words “banana” and "purple" matches these visual bag of concepts better than alternatives.

---

### Meta-Review · Area_Chair_SQNL · 2023-09-18

**Recommendation:** 5

**Metareview:**

This work investigates a reason for the zero-shot failure of large-scale Vision-Language Models (such as CLIP) on compositional VQA questions, showing these models treat the image and the text inputs as two separate "bags of concepts" and attempt to fill in the other missing concept crossmodally, and that this issue is more prevalent in models trained with contrastive loss as opposed to an autoregressive loss. The authors further propose a fine-tuning based solution to mitigate this problem.
The work provides enough experiments using various models (including CLIP, BLIP, BLIP-2 and OFA), and more generally provides some interesting insights and inspiring findings for future work. The paper is well-written in an exposatory style and clearly structured.

---

### Decision · Program_Chairs · 2023-10-07

**Decision:**

Accept-Main

**Comment:**

This work investigates a reason for the zero-shot failure of large-scale Vision-Language Models (such as CLIP) on compositional VQA questions, showing these models treat the image and the text inputs as two separate "bags of concepts" and attempt to fill in the other missing concept crossmodally, and that this issue is more prevalent in models trained with contrastive loss as opposed to an autoregressive loss. The authors further propose a fine-tuning based solution to mitigate this problem.
The work provides enough experiments using various models (including CLIP, BLIP, BLIP-2 and OFA), and more generally provides some interesting insights and inspiring findings for future work. The paper is well-written in an exposatory style and clearly structured.